# Hepatitis C Virus Infection and Intrinsic Disorder in the Signaling Pathways Induced by Toll-Like Receptors

**DOI:** 10.3390/biology11071091

**Published:** 2022-07-21

**Authors:** Elrashdy M. Redwan, Abdullah A. Aljadawi, Vladimir N. Uversky

**Affiliations:** 1Biological Science Department, Faculty of Science, King Abdulaziz University, P.O. Box 80203, Jeddah 21589, Saudi Arabia; lradwan@kau.edu.sa (E.M.R.); aaljaddawi@kau.edu.sa (A.A.A.); 2Therapeutic and Protective Proteins Laboratory, Protein Research Department, Genetic Engineering and Biotechnology Research Institute, City for Scientific Research and Technology Applications, New Borg EL-Arab, Alexandria 21934, Egypt; 3Department of Molecular Medicine and USF Health Byrd Alzheimer’s Research Institute, Morsani College of Medicine, University of South Florida, Tampa, FL 33612, USA

**Keywords:** intrinsically disordered region, intrinsically disordered protein, hepatitis C virus, toll-like receptors, protein–protein interactions, disorder-to-order transition, posttranslational modifications

## Abstract

**Simple Summary:**

Human hepatitis C virus causes contagious liver disease hepatitis C, which is broadly spread around the globe. Infection with this virus causes an inflammation of the liver that leads to both acute and chronic hepatitis, which, if left untreated, often results in the development of serious lifelong illnesses such as liver fibrosis, liver cirrhosis, hepatocellular carcinoma, and end-stage liver disease in humans. Toll-like receptors are a class of pattern recognition receptors that exist in liver parenchymal cells and various immune cells, and play an important role in the liver immune system by initiating multiple cellular downstream pathways in response to the recognition of pathogens such as hepatitis C virus. Proteins from the hepatitis C virus, Toll-like receptors themselves, and proteins from the pathways they activate are multifunctional. Since protein multifunctionality is commonly associated with intrinsic disorder (i.e., lack of stable 3D structures), we used an intrinsic disorder angle to examine the interplay between the hepatitis C virus infection and signaling pathways induced by Toll-like receptors. We show that almost all of these proteins contained noticeable levels of disorder, indicating that intrinsic disorder is prominently utilized in virus–host warfare.

**Abstract:**

In this study, we examined the interplay between protein intrinsic disorder, hepatitis C virus (HCV) infection, and signaling pathways induced by Toll-like receptors (TLRs). To this end, 10 HCV proteins, 10 human TLRs, and 41 proteins from the TLR-induced downstream pathways were considered from the prevalence of intrinsic disorder. Mapping of the intrinsic disorder to the HCV-TLR interactome and to the TLR-based pathways of human innate immune response to the HCV infection demonstrates that substantial levels of intrinsic disorder are characteristic for proteins involved in the regulation and execution of these innate immunity pathways and in HCV-TLR interaction. Disordered regions, being commonly enriched in sites of various posttranslational modifications, may play important functional roles by promoting protein–protein interactions and support the binding of the analyzed proteins to other partners such as nucleic acids. It seems that this system represents an important illustration of the role of intrinsic disorder in virus–host warfare.

## 1. Introduction

### 1.1. Brief Introduction of Human Hepatitis C Virus (HCV)

Human hepatitis C virus (HCV), which causes the contagious liver disease hepatitis C, is one of the best known members of the *Hepacivirus* genus of the *Flaviviridae* family of small enveloped RNA viruses. The capsids of these spherical viruses range in diameter between 40 and 60 nm, and their genomes represent single-stranded positive-sense RNA containing 9.6 to 12.3 kb. In addition to HCV, the *Hepacivirus* genus includes hepatitis G virus (HGV, also known as the GB virus B (GBV-B)), bat hepacivirus, canine hepacivirus, horse hepacivirus, and rodent hepacivirus. Furthermore, there are three other genera in the *Flaviviridae* family, with many viruses from this family predominantly spread by mosquitoes and ticks. In the *Flavivirus* genus, there are almost 70 human and animal viruses, the best-known being Alkhurma hemorrhagic fever virus (AHFV), Dengue virus (DENV), Japanese encephalitis virus (JEV), Kyasanur Forest disease virus (KFDV), Omsk hemorrhagic fever virus (OHFV), West Nile virus (WNV), yellow fever virus (YFV), and Zika virus (ZKV). The *Pegivirus* genus includes GBV-A, GBV-C, and GBV-D (the GB in their names and in GBV-B is related to the initials of the surgeon with acute hepatitis, whose serum, being inoculated into tamarins, induced hepatitis [1,2]). Finally, in the *Pestivirus* genus, one might find atypical porcine pestivirus, Aydin-like pestivirus, bovine viral diarrhea viruses 1 and 2 (BVDV-1 and BVDV-2), border disease virus (BDV), Bungowannah virus, classic swine fever virus (CSFV), Dongyang pangolin virus (DYPV), giraffe pestivirus, Hobi-like pestivirus, pronghorn pestivirus, and rat pestivirus. 

Since the liver parenchymal cells represent the preferential sites of the HCV replication, infection with this virus causes an inflammation of the liver that leads to both acute and chronic hepatitis, which, if left untreated, often results in the development of serious lifelong illnesses such as liver fibrosis, liver cirrhosis, hepatocellular carcinoma, and end-stage liver disease in humans [3]. Although in developed and developing countries, HCV is predominantly transmitted via needle sharing during intravenous drug use and unsafe medical procedures (blood transfusion, organ transplantation, contaminated medical instruments, accidental needle stick injury in health care workers), respectively [4], there are other routes of virus transmission via inadvertent exposure to infected blood such as body modification (tattooing and piercing), sexual intercourse, sharing of personal items, and passage from an infected mother to unborn child (vertical transmission). 

According to the World Health Organization (WHO) [5], HCV chronically infects 58 million people worldwide, with about 1.5 million new infections occurring per year. Furthermore, HCV continues to be a leading cause of liver-related mortality worldwide, causing 333,000, 499,000, and 704,000 deaths in 1990, 2010, and 2013, respectively [6,7,8]. Chronic HCV infection rates are unevenly distributed around the globe. In fact, although the total global HCV prevalence is estimated to be 2.5% (177.5 million of HCV infected adults) [9], and although the prevalence of chronic hepatitis C is generally below 2% in high income countries [10,11] (though even in these countries, prevalence of viremic HCV infection ranges from 0.4% (Austria, Cyprus, Germany, Denmark, France, the UK) to 1.5% (Israel, Italy) [11]); in many low-middle income countries, HCV prevalence exceeds 5% [6]. For example, up to 22% of Egypt’s population is chronically infected with this virus [11,12]. Some other countries with high HCV prevalence (≥5%) are Cameroon at 4.9–13.8% [13], Gabon at 4.9–11.2% [14], Georgia at 6.7% [15], Mongolia at 9.6–10.8% [16,17], Nigeria at 3.1–8.4% [13], Pakistan at 6.8% [18], and Uzbekistan at 11.3% [10].

Several factors define the high variability of the HCV genome. An incomplete list of factors that constantly influence HCV evolution and shape the genome of this virus includes peculiarities of the environment, host genotypic variation, and the lack of a proofreading mechanism during replication [19,20]. As a result, HCV is characterized by the high genetic heterogeneity, and one can find seven major HCV genotypes (1–7) [21] with an average difference of 30% at the nucleotide level [22]. These genotypes are further subdivided into 86 subtypes [23], which are unevenly distributed between genotypes [24], and which, at the nucleotide level, show a difference ranging between 15 and 25% [22]. Similar to the uneven worldwide distribution of global HCV infection, genotypes are distributed disproportionally within the infected population, with the geographic regions being characterized by different relative prevalence of the HCV genotypes. One of the illustrations of such disproportional distribution is given by the United States, where genotypes 1 and 2 are found in 70% and 20% of HCV cases, respectively, with other genotypes each accounting for ~1% [25]. However, the Swat of Khyber Pakhtoonkhaw district of Pakistan was characterized by a very different distribution, where 3a was the most prevalent HCV genotype (34.1%), followed by 2a (8.1%), 3b (7%), and 1a (5.4%) [26].

Furthermore, although genotype 1 is the most common cause of the HCV infection in South America and Europe [3], in South Africa and Hong Kong, the commonly found genotypes are 5 and 6, respectively, whereas HCV-4 is the most common genotype in North Africa and the Middle East [22,27]. Curiously, several specific geographical locations are characterized by unique non-common HCV genotypes. These include Vietnam, where patients are commonly infected with the HCV genotypes 7, 8, and 9 [28,29], and by Indonesian patients with a wide distribution of genotypes 10 and 11 [30]. Finally, the HCV subtypes are also disproportionally distributed among different geographical locations. For example, Japanese patients are preferentially infected by the HCV subtype 1b [31], whereas the most common subgenotypes found in European and U.S. patients are the HCV subtypes 1a and 1b [32,33,34]. Similarly, although HCV subtype 2c is exclusively found in Northern Italy, North America, Europe, and Japan, they are also places where the most commonly found HCV subtypes are 2a and 2b [31,32,33,35]. Another important mechanism allowing HCV to efficiently escape the host immune response and complicating the development of vaccines against this virus is the error-prone RNA polymerase-triggered generation of quasi-species in infected individuals [36].

HCV is an RNA virus with a single-stranded positive-sense RNA genome containing a single 9600-nucleotide-long open reading frame [37] encoding a ~3010-residue-long polypeptide known as the HCV genome polyprotein [38]. Processing of this polyprotein through viral and cellular proteases generates ten HCV proteins with various vital roles in the viral replication and viral particle assembly within the host cell [38]. Some of these proteins such as core or nucleocapsid protein C (p22) and envelope glycoproteins E1 (or gp35) and E2 (or gp70) are structural proteins that assemble to form the viral capsid surrounding and protecting the viral genomic RNA. These structural proteins are released from the HCV genome polyprotein by the host endoplasmic reticulum (ER) signal peptidase(s) [39]. The remaining HCV proteins are non-structural proteins, which are required for viral replication and are released from the polyprotein by the cleavage action of the HCV proteases NS2-3 and NS3-4A [38,39]. These non-structural proteins are a viral channel forming protein p7, transmembrane protein NS2 (p23), protease/RNA helicase NS3 (p70), cofactor NS4A (p8), cofactor NS4b (p27), interferon resisting protein NS5A (p56), and RNA polymerase NS5B (p68) [38]. Importantly, similar to proteins from other viruses, all HCV proteins are multifunctional and are characterized by high binding promiscuity [40,41], which is illustrated by Figure 1, showing the interactions between the HCV and host cell proteins [41].

### 1.2. Introduction of the Protein Intrinsic Disorder Phenomenon

It is recognized that the robustness, multifunctionality, and binding promiscuity of viral proteins can be attributed to their unique structural features and characteristics [42] such as significant enrichment in polar residues and depletion in hydrophobic residues [43], relatively low package density, an increased fraction of residues not involved in secondary structure elements, relatively weak network of inter-residue interactions, lower destabilizing effects of mutations [42], and an abundant presence of partially folded, incompletely ordered, or intrinsically disordered domains and regions that grant structural and functional flexibility to proteins carrying them [44]. This is in line with the current understanding of the protein sequence–function relationship, where protein functionality is not necessarily dependent on a unique 3D-structure, and many biologically active proteins are completely or partially disordered [45,46,47,48,49,50,51,52,53].

**Figure 1 biology-11-01091-f001:**
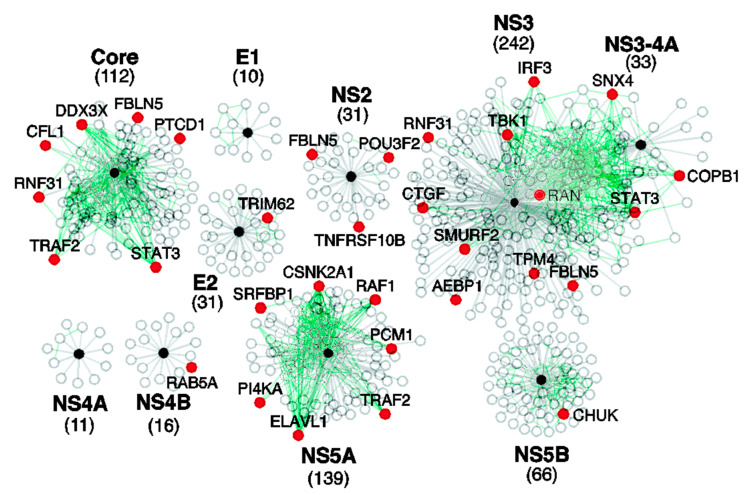
The interactions between HCV and cellular proteins. HCV-human protein–protein interactions (grey lines) were downloaded from the HCVPro database [54] and deduced from references [55,56]. The black spheres represent HCV proteins, the white spheres indicate cellular proteins that bind to HCV proteins, and the red spheres signify human proteins that bind to HCV proteins and were implicated in HCV replication by at least one large-scale siRNA screen. The green lines represent interactions between cellular proteins that bind to HCV proteins. The total number of cellular proteins that bind to each HCV protein is shown in parentheses beneath the protein name. Reproduced from [41] with permission from the Royal Society of Chemistry.

Such functional intrinsically disordered proteins (IDPs) and regions (IDRs) do not have stable 3D structures, being present instead as highly dynamic conformational ensembles [45,47,51,57,58,59,60]. Intrinsic disorder in proteins comes in multiple flavors, and entire protein or protein regions can be disordered to different degrees, existing as collapsed, semi-collapsed, or extended conformational ensembles with molten globule-like, pre-molten globule-like, and coil-like properties [48,50,60,61]. Overall, any protein can be considered as an ensemble of units capable of independent folding (foldons), IDRs undergoing disorder-to-order transitions induced by binding to specific partners (inducible foldons), IDRs with the capability to fold differently being bound to different partners (morphing inducible foldons), permanently disordered regions (non-foldons), regions with perpetually semi-folded conformations (semi-foldons), and ordered regions, the functional activation of which requires order-to-disorder transitions (unfoldons) [60,62,63,64]. This structural heterogeneity of proteins is related to their multifunctionality via the structure–function continuum model, which is based on the “one gene–many proteins–many functions” concept, instead of the classical “lock-and-key” theory rooted in the long-accepted “one gene–one protein–one structure–one function” view [65,66]. In fact, IDPs/IDRs are commonly related to the regulation of cell signaling, are involved in a multitude of recognition events, and play important roles in controlling various pathways [47,50,52,60,67,68,69,70,71,72,73,74,75,76], and thereby complement catalytic and transport activities traditionally attributed to ordered proteins [58,77,78,79,80]. Furthermore, these multifunctional flexible proteins are commonly related to the pathogenesis of various human diseases [81,82].

It is now recognized that IDPs/IDRs are not mere exceptions that are rarely found within the functional universe of ordered proteins, where a unique structure defines the unique function. Instead, they are very abundant in all proteomes analyzed so far [45,46,49,50,53], with the largest variability in the proteome-wide content of disordered residues being found in viral proteomes [83,84]. Recently, the abundance of intrinsic disorder in the completed proteomes of several human HCV genotypes (such as 1a, 1b, 1c, 2a, 2b, 2c, 2k, 3a, 3b, 3k, 4a, 5a, 6a, 6b, 6d, 6g, 6h, and 6k) was evaluated using a wide spectrum of bioinformatic techniques [41]. This analysis was complemented by the investigation of the peculiarities of disorder distribution within the individual HCV proteins, which helped establish a potential conjunction between the structural disorder and functions of the ten HCV proteins [41]. This analysis revealed that “intrinsic disorder or increased flexibility is not only abundant in these proteins, but is absolutely necessary for their functions, playing a crucial role in the proteolytic processing of the HCV polyprotein, the maturation of the individual HCV proteins, and being related to the posttranslational modifications of these proteins and their interactions with DNA, RNA, and various host proteins” [41]. In line with these previous studies, our analysis (see below) showed that HCV contains four ordered proteins (E1, E2, NS2, and NS4B), five moderately disordered proteins (p7, NS3, NS4A, and NS5B), and two highly disordered proteins (core and NS5A).

## 2. TLRs, Important Players of the Liver Immune System

Toll-like receptors (TLRs), a class of pattern recognition receptors, exist in liver parenchymal cells and various immune cells, and serve as important members of the liver immune system [20], playing a number of vital roles in both inflammatory and infectious diseases [85,86]. In fact, these receptors are known to improve the overall efficiency of the immune response, being able to mediate innate immunity and induce acquired immunity. In innate immunity, the main functions of TLRs are related to the recognition of danger-associated molecular patterns (DAMPs) and pathogen-associated molecular patterns (PAMPs) [87]. Recognition of PAMPS and DAMPs by TLRs initiates a cascade of events that eventually results in the killing of a microbe. The corresponding events range from the recruitment of phagocytes to the site of infection to the initiation of the expression of chemokines and cytokines acting as inflammatory mediators [88,89].

There are ten TLRs (TLR1 through TLR10) in humans [90,91,92,93,94], which are differently distributed within a cell and can be further grouped based on their cellular location, with TLR1, TLR2, TLR4, TLR5, TLR6, and TLR10 being found on the plasma membrane, and TLR3, TLR7, TLR8, and TLR9 being located on the endosomal membranes [20,93,94]. Importantly, this diversity of human TLRs is determined by the need to recognize a multitude of the PAMPs from bacteria, fungi, protozoa, and viruses [89,90,93], with each TLR being able to sense a specific set of PAMPs. Here, the nucleic components of pathogens are mainly recognized by the endosomal TLRs, where double-strand RNA, single-stranded RNA, and CpG DNA are sensed by TLR3, TLR7/TLR8, and TLR9, respectively [89,90,93,94,95]. In response to their corresponding PAMPs, these TLRs stimulate the production of type 1 interferons (IFN1α and IFN1β) [96]. As far as extracellular TLRs are concerned, they can sense a broad range of PAMPs, with TLR5 sensing flagellin and TLR4 recognizing many different PAMPs such as lipopolysaccharides and taxol [89,91]. PAMP-induced activation of these TLRs at the plasma membrane triggers the activation of nuclear factor-κB (NF-κB) and other transcription factors that lead to the expression of several proinflammatory cytokines [97].

TLRs are involved in the interaction with each other (in fact, activation of these receptors is associated with their homo- and heterodimerization) and a multitude of human proteins. This is illustrated by Figure 2, which shows the internal and external protein–protein interaction (PPI) networks of these proteins. This analysis revealed that TLRs form a densely connected network, where, on average, each protein interacts with at least eight partners. The most connected are TLR1, TLR7, and TLR9, which are expected to interact with all of the remaining TLRs, and the least connected is TLR10, interacting with TLR1, TLR2, TLR3, TLR5, TLR7, and TLR9 (see Figure 2A). As a group, TLRs form an immense PPI network, where one can find 459 proteins (see Figure 2B). In line with these observations, STRING-based analysis of the interactivity revealed that individual TLRs are engaged in multiple PPIs (see Appendix A and corresponding plots in the Appendix A).

A detailed description of the functional peculiarities of TLRs was outside the scope of this article, and interested readers can find related information in numerous reviews dedicated to this important family of pattern recognition receptors (e.g., see [100,101,102,103,104,105,106,107,108,109,110,111,112,113,114,115,116,117]). However, the data presented here indicate that TLRs belong to the category of multifunctional proteins. It is likely that this multifunctionality is somehow encoded in the structures of these proteins.

## 3. Structure and Intrinsic Disorder in TLRs and Major Players in the TLR-Triggered Cellular Pathways

TLRs are type I (single-pass) transmembrane glycoproteins consisting of three functional domains such as an N-terminal extracellular (or extra-endosomal) leucine rich repeat domain (LRR domain also known as ectodomain) responsible for the recognition of PAMPs and DAMPs, a transmembrane domain (containing an α-helix, which is 21-residue-long in all human TLRs) responsible for the membrane anchoring of TLRs needed for the maintenance of their functional topologies, and a C-terminal intracellular (intraendosomal) Toll/interleukin-1 receptor (TIR) domain with a crucial role in transmitting the extracellular/extraendosomal signals into the cell or endosome [118,119,120]. Ligand binding to ectodomains induces their homo- or heterodimerization, which triggers dimerization of the TIR domains [118]. Activated TLR homo- and heterodimers are characterized by a strikingly similar “M”-like shape, where the C-terminal regions of the ectodomains converge in the middle [118]. This ligand-induced dimerization triggers the recruitment of various adaptor proteins to the intracellular TIR domains of TLRs, thereby initiating corresponding signaling pathways [121].

The LRR domain contains a series of 16–28 repeated LRR modules [122], with each LRR module being 20–30 residues in length and containing a conserved “LxxLxLxxN” motif and a variable part [123,124]. The conserved sequence patterns in the LRR modules define the capability of LRR proteins to fold into the unique horseshoe-like solenoid shape, where the inner concave surface is formed by the “LxxLxLxxN” motives organized in parallel β strands, and the outer convex surface originates from the variable parts of the LRR modules forming α-helices, β-turns, and/or loops [123,124]. Not all LRR modules are made equal, and some LRR proteins contain N- and C-terminally located LRRNT and LRRCT modules that often possess clusters of cysteine residues forming disulfide bridges, but not including LRR motives [123,124].

Human TLR1, TLR2, TLR3, TLR4, TLR5, TLR6, TLR8, and TLR10 contain 19, 19, 22, 18, 19, 19, 23, and 15 LRR modules, and their LRRCT modules are 55-, 55-, 54-, 51-, 53-, 55-, 53-, and 55-residues-long. The 28-residue-long LRRNT module is present only in TLR3, whereas TLR7 and TLR9, being among the longest human TLRs (see Appendix A), contain 27 and 24 LRR modules, respectively, but do not have LLRNT and LLRCT modules. Based on their structural patterns and sequences, TLRs belong to the “typical” subfamily of the LRR family proteins that contain 24-residue-long LRR modules with the conserved motif “xLxxLxxLxLxxNxLxxLPxxxFx” and that, in the convex region of their horseshoe-like structure, contain parallel 3_10_ helices [122,123,124]. A more detailed description of the structural peculiarities of the LRR domains can be found elsewhere [118,125].

The TIR domains are ~150-residue-long intracellular/intraendosomal TIR domains of TLRs characterized by the common fold, where one can find five α-helices surrounding a five-stranded β-sheet [118]. Based on the results of the mutational and molecular dynamic simulation analyses, it was concluded that the loop connecting the second β-strand and the second α-helix (so-called BB loop) is crucial for the TIR dimerization and/or recruitment of specific adaptor proteins [118,126] such as MyD88, MAL (also known as TIRAP), TRIF, and TRAM [121]. Importantly, since these adaptor proteins also possess TIR domains, the activation of TLR signaling is critically dependent on TIR–TIR interactions between the receptor–receptor, receptor–adaptor, and adaptor–adaptor [127].

Although ordered ectodomains and TIR domains of TLRs are well-studied, there is almost no information on the prevalence and potential functionality of intrinsically disordered and flexible regions in these proteins. To fill this gap, we conducted a multifactorial analysis of the intrinsic disorder predisposition of these important proteins. This analysis revealed that human TLRs contain multiple IDRs, some of which can be relatively long. For example, the longest IDRs in TLR3 and TLR8 have 58 and 59 residues, respectively, the longest IDRs in TLR1 and TLR2 are 39-residues-long, whereas TLR7 has two 33-residue-long IDRs (see Appendix A and the corresponding plots in Appendix A). Overall, Appendix A shows that, based on their intrinsic disorder content, human TLRs can be arranged as follows: TLR5 < TLR6 < TLR10 < TLR4 < TLR9 < TLR1 < TLR7 < TLR3 < TLR2 < TLR8. The first four TLRs in this list are expected to be mostly ordered, whereas the remaining six TLRs are predicted as moderately disordered (see Appendix A).

Figure 3 provides the results of the global analysis of the intrinsic disorder predisposition of human TLRs and some major members of the TLR-based signaling cascades. In addition to the utilization of protein disorder status classification based on their PPIDR values (see above), proteins can be classified using their levels of average disorder score (ADS), as highly ordered (ADS < 0.15), moderately disordered (ADS between 0.15 and 0.5), and highly disordered (ADS ≥ 0.5). Figure 3A shows the correlation between the ADS and PPIDR values for the human proteins analyzed in this study.

This analysis revealed that no TLRs can be classified as very structured, since none of these proteins are located within the dark blue region, and only four proteins (TLR4, TLR5, TLR6, and TLR10) can be considered as mostly disordered, with the remaining TLRs being moderately disordered. Further analysis of the global intrinsic disorder predisposition of these proteins is illustrated by Figure 3B, which shows the corresponding charge-hydropathy (CH)–cumulative distribution function (CDF) plot. Here, proteins are classified based on their position within the quadrants of the CH–CDF phase space, where ordered proteins are located within the lower-right quadrant, native molten globules and/or hybrid proteins containing sizable levels of order and disorder are grouped within the lower-left quadrant, and native coils or native pre-molten globules (i.e., highly disordered proteins with extended disorder) are found within the upper-left quadrant [138]. As per the results of this analysis, all TLRs were grouped within the lower-right quadrant, confirming that these proteins were mostly ordered.

To show the peculiarities of the structure and disorder distribution within the human TLRs, we modeled their structures by AlphaFold [142] and also used the D^2^P^2^ database to generate functional disorder profiles of these proteins [143]. Figure 4 gives an illustration of this analysis, showing the corresponding results for TLR5 and TLR8 as the most ordered and most disordered human TLRs. Analogous information for other TLRs is presented in the Appendix A.

Figure 4 shows that proteins that contain both disordered regions and disorder or structural flexibility might have some functional implications (e.g., be involved in posttranslational modifications). The involvement of disorder/flexibility in PTMs is supported by the fact that in TLR5, Asn residues 37, 46, 245, 342, 422, 595, and 598, which are subjected to N-linked glycosylation [144], are characterized by the local disorder scores of 0.051, 0.259, 0.279, 0.146, 0.181, 0.097, and 0.150, respectively, and the phosphorylatable residues Tyr798 [145] and Ser805 [146] showed local disorder scores of 0.411 and 0.328, respectively. In TLR8, the situation was even more dramatic, as Asn residues 29, 42, 80, 88, 115, 160, 247, 285, 293, 358, 362, 395, 416, 443, 511, 546, 582, 590, 640, 680, and 752, which were subjected to glycosylation, showed local disorder scores of 0.542, 0.620, 0.368, 0.439, 0.542, 0.234, 0.198, 0.133, 0.368, 0.344, 0.344, 0.254, 0.150, 0.706, 0.234, 0.422, 0.150, 0.247, 0.235, 0.082, and 0.672, respectively. This is in line with previous observations indicating that high levels of disorder or structural flexibility in proteins often serve as a signal for a variety of posttranslational modifications [147,148,149].

At the next stage, we looked at the intrinsic disorder status of major players of the TLR-initiated signaling cascades. Results of the corresponding analyses are summarized in Appendix A, Figure 3, and the corresponding plots in the Appendix A and show that all 41 proteins are expected to contain noticeable levels of intrinsic disorder. In fact, 13 of these proteins are predicted to have more than 50% disordered residues, and another 16 proteins possessed PPIDR values exceeding 25%, whereas disorder content in the remaining 12 proteins ranged from 10.62% (serine/threonine-protein kinase TBK1; TANK-binding kinase 1) to 24.56% (inhibitor of nuclear factor-κB (NF-κB) kinase subunit alpha, IKKα). Similarly, in the CH–CDF plot (see Figure 3B), one of the members of the TLR signaling cascades was predicted as highly disordered, 17 were expected to have a molten globular or hybrid structure, and 23 were mostly ordered.

**Figure 4 biology-11-01091-f004:**
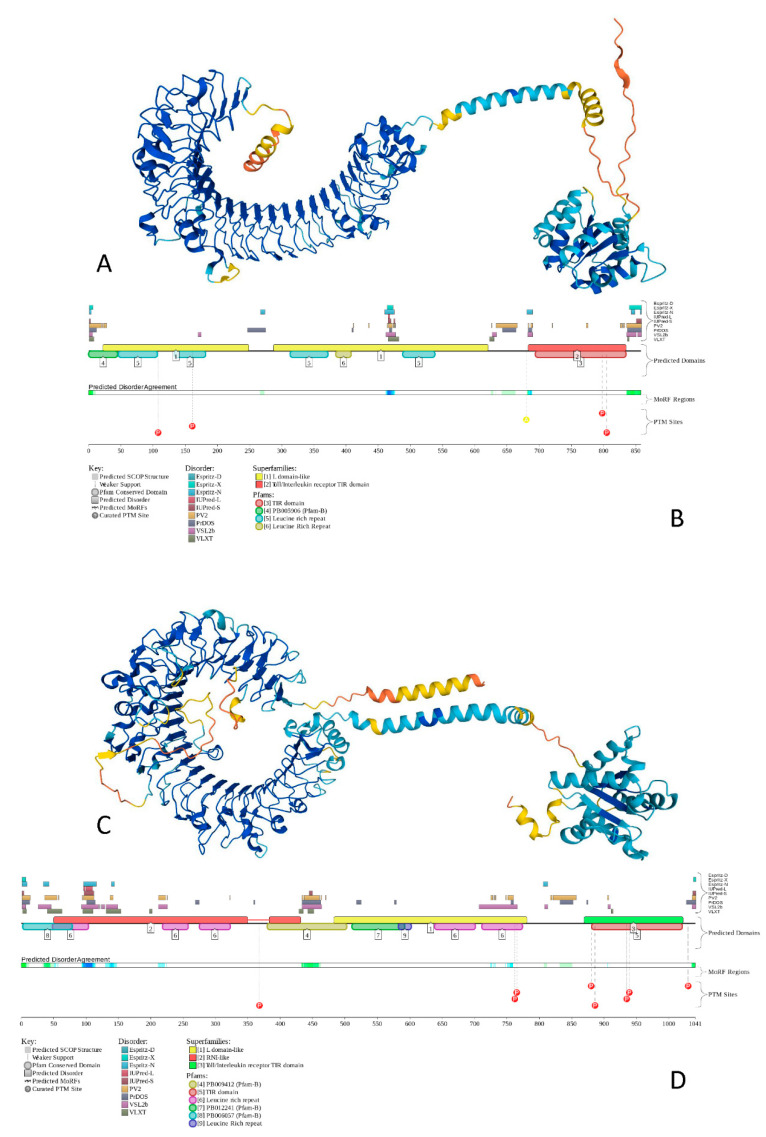
The structure and functional disorder of the most ordered (TLR5, plots (**A**,**B**)) and most disordered (TLR8, plots (**C**,**D**)) human TLRs. Structures of these proteins (plots (**A**,**C**)) were modeled using AlphaFold [142], which is recognized now as a highly accurate tool suitable for large-scale structure prediction [150]. The modeled structures of all of the query proteins (HCV proteins, human TLRs and 41 major players of TLR-triggered downstream path-ways) are shown in the Appendix A. Structural elements in these structures are colored based on the confidence of the structure prediction by AlphaFold, where dark blue and cyan segments correspond to structures predicted with high to very high confidence, whereas yellow and orange segments showed structures predicted with low to very low confidence, and which are expected to be unstructured in isolation. Functional disorder profiles (plots (**B**,**D**)) for these proteins were generated using the D^2^P^2^ platform (http://d2p2.pro/ accessed on 7 July 2022), which is a database of the predicted disorder for proteins from completely sequenced genomes [143]. Here, the outputs of IUPred [133], PONDR^®^ VLXT [129], PrDOS [151], PONDR^®^ VSL2B [130,152], PV2 [143], and ESpritz [153] were used to show the disorder predispositions. Consensus between these nine disorder predictors is shown by the blue-green-white bar, whereas the location of various posttranslational modifications (PTMs) is shown by differently colored circles. The platform also shows the positions of the functional SCOP domains [154,155] predicted by the SUPERFAMILY predictor [156]. Positions of these functional domains are shown below the outputs of nine disorder predictors. The functional disorder profile also includes information on the location of the predicted disorder-based binding sites (MoRF regions) identified by the ANCHOR algorithm [157] and various PTMs assigned using the outputs of the PhosphoSitePlus [158]. The functional disorder profiles of all query proteins are shown in the Appendix A.

The most disordered TLR signaling proteins with a PPIDR exceeding 70% were the inhibitor of the nuclear factor-κB (NF-κB) kinase subunit gamma (IKKγ; PPIDR = 98.57%; note that IKKγ is commonly known as NF-kappa-B essential modulator NEMO), two members of the AP-1 complex, protein c-FOS (PPIDR = 83.42%) and transcription factor Jun (PPIDR = 82.78%), and TRAF family member-associated NF-kappa-B activator (TANK, PPIDR = 70.35%).

These four proteins were selected for more in-depth disorder analysis, which is summarized in Figure 5, Figure 6 and Figure 7. Figure 5 represents the model 3D-structures generated for these proteins by AlphaFold and clearly shows that none of them are expected to have a compact globular structure.

In fact, C-FOS and Jun are mostly disordered, with each showing a long α-helix related to the formation of the coiled-coil structure in the C-FOS-Jun complex. IKKγ is predicted to have a V-shaped structure with two very long α-helices, whereas TANK, also being highly disordered, is expected to have two disjoined α-helices. Note that none of these proteins have a hydrophobic core, and therefore their long α-helices are unlikely to be stable in the unbound state. Figure 6 represents the functional disorder profiles of human IKKγ, c-FOS, Jun, and TANK generated by the D^2^P^2^ platform [143] and shows that these four proteins are massively disordered and densely decorated by multiple various PTMs. Furthermore, they include numerous molecular recognition features (i.e., intrinsically disordered regions that are expected to undergo disorder-to-order transitions at the interaction with specific binding partners). Such features are commonly found within the IDRs of many proteins, where they were shown to play crucial roles in protein–protein interactions, potentially initiating early steps in molecular recognition [47,48]. IKKγ, c-FOS, Jun, and TANK have 6, 8, 8, and 9 MoRFs that cover 15.75%, 29.21%, 48.04%, and 22.35% of their entire sequences, respectively, indicating that very significant parts of these proteins are involved in molecular recognition.

Furthermore, MoRFs predicted in c-FOS (residues 180–212) and Jun (residues 241–250/252–284) overlap with the long helical segments predicted in these proteins by AlphaFold (residues 134–203 in c-FOS and 247–317). Similarly, two long helical segments of TANK (residues 5–69 and 130–165) included the predicted NoRFs (residues 12–21/54–65 and 142–159). These observations suggest that the indicated MoRFs can potentially adopt an α-helical structure at binding to their partners.

The fact that significant parts of these four proteins are predicted to be engaged in molecular recognition and undergo binding-induced disorder-to-order transition is further supported by Figure 7, which shows dense PPIs predicted for these four proteins by STRING. Each of these proteins represents a highly connected hub, indicating the heavy use of intrinsic disorder for PPIs. This is in line with previous studies that showed that intrinsic disorder is crucial for the functionality of hubs [52,159,160,161,162,163,164].

It should be noted that these four proteins are not an exception, and all other participants of TLR signaling analyzed in this study contain multiple MoRFs and act as hubs in densely packed PPI networks (see the Appendix A for this information).

To conclude this section, Figure 8 schematically represents the TLR-related signaling pathways [20,86,165]. Except for TLR3, which exclusively uses TRIF (Toll/IL-1 receptor domain-containing adaptor inducing IFN-β; PPIDR = 66.29%) as a mediator of downstream signaling, the downstream signaling pathways of all other TLRs are mediated by the TIR domain-containing adaptor MyD88 (myeloid differentiation factor, also known as TIR domain-containing adapter protein TIRAP, PPIDR = 19.93%). TLR4 may initiate both the MyD88-dependent and MyD88-independent pathways. The activation of TLR7/TLR8, TLR9, TLR10, and TLR10/TLR2 leads to the direct binding of their TIR domains to the MyD88 TIR domain. On the other hand, interaction of the TLR1/TLR2, TLR2/TLR6, and TLR4 with MyD88 is bridged by the junction protein MAL (MyD88 adapter-like, PPIDR = 42.53%) [166,167].

MyD88 binding to the activated TLRs directly or via MAL initiates the MyD88-dependent downstream signaling pathways [166,167]. Here, MyD88 interacts and activates the IL-1 receptor-associated kinases IRAK1 (PPIDR = 52.67%) or IRAK4 (PPIDR = 33.48%). Activated IRAKs form a complex with TRAF6 (tumor necrosis factor (TNF) receptor-associated family 6, PPIDR = 23.37%). This leads to the activation of TAK1 (transformation growth factor-β (TGF-β)-activated kinase, PPIDR = 57.10%), which activates the IKKα/IKKβ/IKKγ complex (inhibitors of nuclear factor-κB (NF-κB) kinase α, β, and γ, PPIDR = 24.56%, 35.45%, and 98.57%, respectively) and MAPKs (mitogen-activated protein kinases 1–15, whose PPIDR values range from 13.89% in MAPK14 to 63.24% in MAPK7).

Note that MAPK8, MAPK9, and MAPK10 are also known as c-Jun N-terminal kinases 1, 2, and 3, JNK1, JNK2, and JNK3 with PPIDs of 27.63%, 27.36%, and 26.72%. MAPK3 (PPIDR = 22.95%) and MAPK1 (PPIDR = 16.67%) are also known as extracellular signal-regulated kinases 1 and 2 (ERK1 and ERK2), respectively. Finally, MAPK11 (p38-beta; PPIDR = 21.15%), MAPK12 (p38-gamma; PPIDR = 24.52%), MAPK13 (p38-delta; PPIDR = 30.68%), and MAPK14 (p38-beta; PPIDR = 13.89%) represent four p38 MAPKs that play important roles in the cascades of cellular responses evoked by extracellular stimuli. Activated MAPKs trigger the nuclear translocation of the activated protein-1 (AP-1) complex containing c-Jun (PPIDR = 82.78%) and c-FOS (PPIDR = 83.42%). The IKKα/IKKβ complex releases the NF-kappa-B inhibitor alpha (IkBa, PPIDR = 39.75%) from the NF-κB (a complex containing subunits p65 and p50 with PPIDRs of 64.61% and 33.16%, respectively) and triggers its nuclear translocation. In the nucleus, AP-1 and NF-κB initiate biosynthesis of the proinflammatory cytokines [166,167]. Alternatively, activated TRAF6 can trigger the IRF5 (interferon regulatory factor 5; PPIDR = 43.17%) pathway, leading to the nuclear internalization of this factor, where it activates the expression of type I interferons (IFNs) IFNA (PPIDR = 31.75%) and INFB (PPIDR = 11.76%) and inflammatory cytokines [168,169,170,171,172].

TL3 and TL4 can initiate MyD88-independent downstream signaling pathways, with TLR4 being the only TLR that can trigger both MyD88-dependent and -independent pathways. In the MyD88-independent pathway, TLR4 complexed with CD14 (cluster of differentiation 14; PPIDR = 25.87%), which is a high affinity, horse shoe-shaped, glycosylphosphatidylinositol-(GPI-) anchored membrane protein and is internalized into the endosomes, where TRIF-related adaptor molecule (TRAM; PPIDR = 50.21%) binds to TLR4 and ensures the recruitment of TIR domain-containing adaptor-inducing interferon-β (TRIF; PPIDR = 66.29%) to TLR4 [173,174]. Then, TRAF family member-associated NF-κB activator, (TANK-binding kinase-1 (TBK-1; PPIDR = 10.62%) and the inhibitor of nuclear factor-κB (NF-κB) kinase subunit epsilon (IKKε; PPIDR = 23.88%) are activated, leading to the phosphorylation and activation of interferon regulatory factor 3 (IRF3; PPIDR = 37.94%), which enters the nucleus and initiates the transcription of anti-inflammatory cytokines [175]. Furthermore, in this pathway, TRIF can activate NF-κB and promote its nuclear translocation through receptor-interacting protein 1 (RIP1; PPIDR = 47.84%) and transformation growth factor-β (TGF-β)-activated kinase (TAK1; PPIDR = 57.10%) or TRAF6 [176]. It can also act via the activation of MAPKs to induce AP-1, thereby triggering a proinflammatory response [177].

The internalization of viral PAMPs leads to the activation of the TLR3, TLR7/8, and TLR9 in the endosomes. The activation of TLR3 is associated with the initiation of the MyD88-independent pathways. Similar to TLR4, TLR3 interacts with TRIF, and this leads to the IKKε/TBK-1-driven activation of IRF3. Although TLR7, TLR8, and TLR9 are on the MyD88-dependent pathway, MAL is not needed for their interaction with MyD88. After MyD88 binding, they activate IRAKs that bind to TRAF6, leading to IRF5 and IRF7 (PPIDR = 58.08%) activation and translocation to the nucleus, where these interferon regulatory factors trigger the transcription of anti-inflammatory cytokines.

Finally, the role of TRL10 also needs to be addressed, which, despite being discovered in 2001 [178], is considered to be an orphan receptor whose ligands and functions are poorly known [86,179,180]. Although this receptor can homodimerize and form heterodimers with TLR1, TLR2, and TLR6 [181,182,183,184], the functional significance of the resulting complexes and TLR10 itself remains mostly unknown [182]. It was shown that the TLR10/TLR2 heterodimer and TLR10/TLR10 can bind MyD88, but they are incapable of NF-κB activation [185]. It seems that TLR10 serves as an inhibitor of the MyD88 dependent and independent pathways [86,178].

The data presented in this section are summarized in Figure 8, showing the proteins involved in these TLR pathways as being colored based on their intrinsic disorder status, with highly ordered, moderately disordered, and highly disordered proteins being shown by the blue, pink, and red colors, respectively. With the exception of four TLRs (TLR4, TLR5, TLR6, and TLR10), all the proteins in these networks were either moderately or highly disordered. This once again emphasizes the importance of intrinsic disorder for the regulation and control of these crucial pathways.

## 4. TLRs in HCV Infection

Although host cells contain several pattern recognition receptors that can recognize HCV such as C-type lectin receptors (CLRs), cytosolic DNA sensors (CDs), NOD-like receptors (NLRs), RIG-I-like receptors (RLRs), and Toll-like receptors (TLRs) [36], TLRs deserve special attention due to their capability to rapidly establish antiviral response and serve as important players in the process of HCV infection [20]. This is because TLRs can recognize several viral PAMPs and efficiently bind to various HCV structures [20]. A detailed description of the roles of HCV proteins in altering and modulating the TLR-based responses can be found elsewhere [20]. The section below provides a brief overview of this phenomenon from the angle of intrinsic disorder.

Figure 9 shows that half of the HCV proteins is tightly related to the TLRs. For example, interactions of the HCV core (PPIDR = 68.18%), NS3 (PPIDR = 22.03%), and NS5A (PPIDR = 47.99%) proteins with TLR1/TLR2 and TLR2/TLR6 activate the downstream TLR2-specific signaling processes in the MyD88-dependent manner (see Figure 8). Here, the TLR2-specific intracellular pathway is utilized by the HCV core and NS3 proteins to activate innate immune and inflammatory cells [186]. Both HCV core and NS3 proteins can be recognized by the TLR2/TLR1 and TLR2/TLR6 heterodimers [187]. Importantly, in chronic HCV infection, the HCV-triggered activation of NF-κB (a complex containing subunits p65 and p50 with PPIDRs of 64.61% and 33.16%, respectively) and AP-1 (a complex of c-Jun (PPIDR = 82.78%) and c-FOS (PPIDR = 83.42%)) transcription factors, and the phosphorylation of JNKs (JNK1, JNK2, and JNK3 with PPIDs of 27.63%, 27.36%, and 26.72%) via the MyD88-driven recruitment of IRAK1 (PPIDR = 52.67%), IRAK4 (PPIDR = 33.48%), and TRAF6 (PPIDR = 22.37%) is believed to be linked to hepatocyte damage, as both hepatocyte metabolism and inflammation/injury/fibrosis are related to the TLR-activated JNKs [188,189].

As was discussed above, TLR4 can activate downstream pathways both in the MyD88-dependent and MyD88-independent manner. Although TLR4 activation is typically induced by the lipopolysaccharides (LPSs), HCV NS5A can bind to TLR4 directly, without LPS stimulation, thereby promoting the MyD88-independent activation of the downstream molecules through the pathways related to TRIF (PPIDR = 66.29%) (see Figure 8). This NS5A-TLR4 binding on monocytes leads to an imbalance of inflammatory cytokines enhancing the IL-10 production and decreasing the IL-12 production [191]. This inflammatory cytokine imbalance inhibits the virus-killing effect of the natural killer (NK) cells, thereby helping HCV to evade immune surveillance [191]. Furthermore, NS5A was shown to upregulate the TLR4 expression in the peripheral blood mononuclear cells and B cells, leading to the enhanced production of IFN-β and IL-6 [192]. On the other hand, in hepatocytes, HCV NS5A was shown to downregulate TLR4 expression, thereby inhibiting the LPS-mediated apoptosis of hepatocytes [193].

In addition to the effects associated with the activation of TLRs by HCV proteins, NS3 and NS4A proteins can interact with the downstream molecules TBK1 (PPIDR = 10.62%) and TRIF (PPIDR = 66.29%), thereby distorting the TLR3-based MyD88-independent pathways, leading to the initiation of the IFN production. Here, the TLR3 adaptor protein TRIF is proteolytically degraded by HCV NS3/NS4A, which impairs the IRF3 (PPIDR = 37.94%)-induced production of INF, whereas the direct NS3 binding to TBK1 damages thee activation of IRF3 and upregulation of IFN-β (PPIDR = 11.76%) [20,194,195]. It was also shown that the IFN-β production can be promoted by NS5B (PPIDR = 15.40%), which may synthesize a nonspecific double-stranded RNA (dsRNA) as a TLR3 ligand, whereas the activities of NS4A (PPIDR = 18.52%), NS4B (PPIDR = 9.20%), and NS5A (PPIDR = 47.99%) can inhibit this pathway by altering the dsRNA synthesis or distorting the TLR3-dsRNA recognition [196].

TLR7/8 can be activated by the HCV single-stranded RNA. It was shown that in the peripheral blood monocytes of the HCV-infected patients, the TLR7 expression was noticeably reduced due to the TLR7 gene distortion and instability of the TLR7 mRNA [197]. Furthermore, NS5A can inhibit the production of some inflammatory factors by binding to the MyD88 death domain, thereby suppressing TLR7/8 signaling [198,199] and affecting other MyD88-dependent pathways downstream of TLRs [20].

Finally, since TLR9 can only bind to DNA, the HCV single-stranded RNA cannot activate this receptor [200]. However, TLR9 can still affect HCV infection via apoptotic cell DNA [201]. Furthermore, similar to TLR7, the levels of the TLR9 mRNA and protein are downregulated in the peripheral blood mononuclear cells from HCV infected patients, being negatively correlated with serum viral copies [202].

## 5. Conclusions

In this article, we analyzed the prevalence and potential functional roles of intrinsic disorder in HCV proteins, TLRs, and some major players of the TLR-induced downstream pathways. To the best of our knowledge, this work, being a compilation of the state-of-the-art and common knowledge on HCV infection and pathophysiological conditions, represents a first systematic study of the intrinsic disorder-based interplay between HCV, TLRs, and TLR-induced downstream pathways. We confirmed that HCV contains four ordered proteins (E1, E2, NS2, and NS4B), five moderately disordered proteins (p7, NS3, NS4A, and NS5B), and two highly disordered proteins (core and NS5A). We also showed that based on their intrinsic disorder content, human TLRs can be arranged as follows: TLR5 < TLR6 < TLR10 < TLR4 < TLR9 < TLR1 < TLR7 < TLR3 < TLR2 < TLR8. The first four TLRs in this list are expected to be mostly ordered, whereas the remaining six TLRs were predicted as moderately disordered. Analysis of the 41 TLR signaling-related proteins revealed that they contained high levels of intrinsic disorder that ranged from 10.62% in TBK1 to 98.75% in IKKγ (NEMO). Of these proteins, 13 are expected to have more than 50% disordered residues, 16 proteins possess PPIDR values between 25% and 50%, and the remaining 12 proteins have disorder content that ranges from 10.62% (TBK1) to 24.56% (IKKα). In all HCV, TLRs, and TLR signaling proteins analyzed in this study, disordered residues are assembled into the functional IDRs, which serve either as a signal for a variety of posttranslational modifications or are used for protein–protein interactions, often being capable of undergoing the binding-induced folding at interaction with specific partners.

The fact that most of these proteins have intrinsic disorder, are capable of undergoing functional disorder-to-order transitions, and can be subjected to multiple PTMs indicates that each of them, despite being encoded by a single gene, exists as a dynamic ensemble of structurally and functionally distinct protein molecules, known as proteoforms [65,203]. Therefore, the functionality of these proteins can be described within the frames of the “protein structure–function continuum” model, where the structure of a protein represents a highly dynamic conformational ensemble containing multiple proteoforms that have different structural features and might have various functions [204,205,206]. Such disorder-based structural and functional heterogeneity of HCV proteins, TLRs, and proteins from the TLR pathways are important for a better understanding of both the HCV pathogenesis and the innate immune response to the HCV infection.

These observations are in line with the well-known association between the protein intrinsic disorder and pathogenesis of various human diseases [81,207,208,209,210,211,212]. They also agree with the results of the comprehensive bioinformatics analyses of the prevalence of protein intrinsic disorder in various viruses such as human papillomaviruses (HPVs) [213], HCV [41], influenza viruses [214], HIV-1 [215], Dengue virus [216], respiratory syncytial virus (RSD) [217], Zika virus [218,219], Chikungunya virus [220,221], rotavirus [222], Japanese encephalitis virus [223], SARS-CoV-2 [224,225], human SARS and bat SARS-like coronaviruses [224], Middle East respiratory syndrome MERS coronavirus [226,227], and Chandipura virus [228] as well as in the interactomes of HPV [229] and HCV [40]. They are also supported by the results of the evaluation of intrinsic disorder in proteins involved in innate anti-viral immunity [230] and the comprehensive bioinformatics analysis of the global prevalence of intrinsic disorder in 6108 viral proteomes [231]. Our study emphasizes the importance of the systematic analysis of intrinsic disorder for a better understanding of the pathogenesis of viral infections. Furthermore, this work provides important information that can be utilized in the future development of the novel therapy against HCV hepatitis.

## Figures and Tables

**Figure 2 biology-11-01091-f002:**
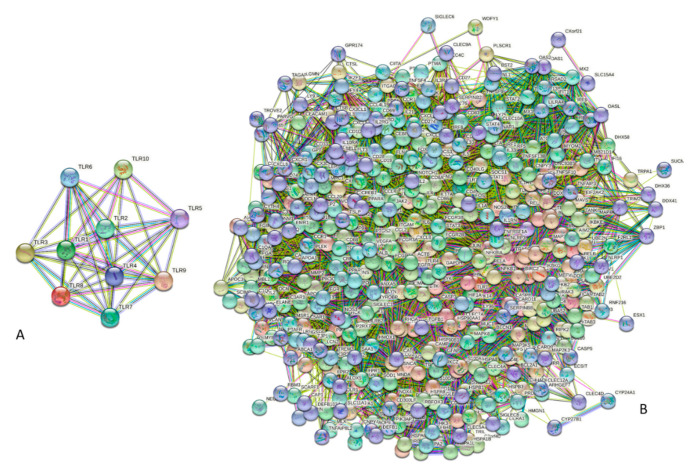
The internal (**A**) and external PPI networks (**B**) of human TLRs generated by STRING (Search Tool for the Retrieval of Interacting Genes, http://string-db.org/ accessed on 7 July 2022) designed to show the predicted and experimentally-validated interactions of a protein of interest [98]. In the internal network, TLRs are connected by 41 interactions, where each TLR, on average, interacts with eight partners. In the external PPI network, 10 TLRs interact with 449 partners, which are connected by 11,715 binary PPIs. The average node degree of these network is 51 (i.e., each member of this network is expected to interact with 51 partners). Amino acid sequences of the HCV polyprotein genotype 1a (isolate H77), mature individual HCV proteins, human TLRs, and 41 major players of TLR-triggered downstream pathways were retrieved from UniProt (https://www.uniprot.org/ accessed on 7 July 2022) [99]. Corresponding information is included in the Appendix A. Search Tool for the Retrieval of Interacting Genes; STRING, http://string-db.org/ accessed on 7 July 2022 [98], was used to obtain information on the interactability of HCV proteins, human TLRs, and 41 major players of TLR-triggered downstream pathways. The STRING output represents a network of predicted and experimentally-validated protein–protein interactions using seven types of evidence such as co-expression evidence (black line), co-occurrence evidence (blue line), neighborhood evidence (green line), database evidence (light blue line), experimental evidence (purple line), fusion evidence (red line), and text mining evidence (yellow line) [98]. The protein–protein interaction networks of all query proteins are shown in the Appendix A.

**Figure 3 biology-11-01091-f003:**
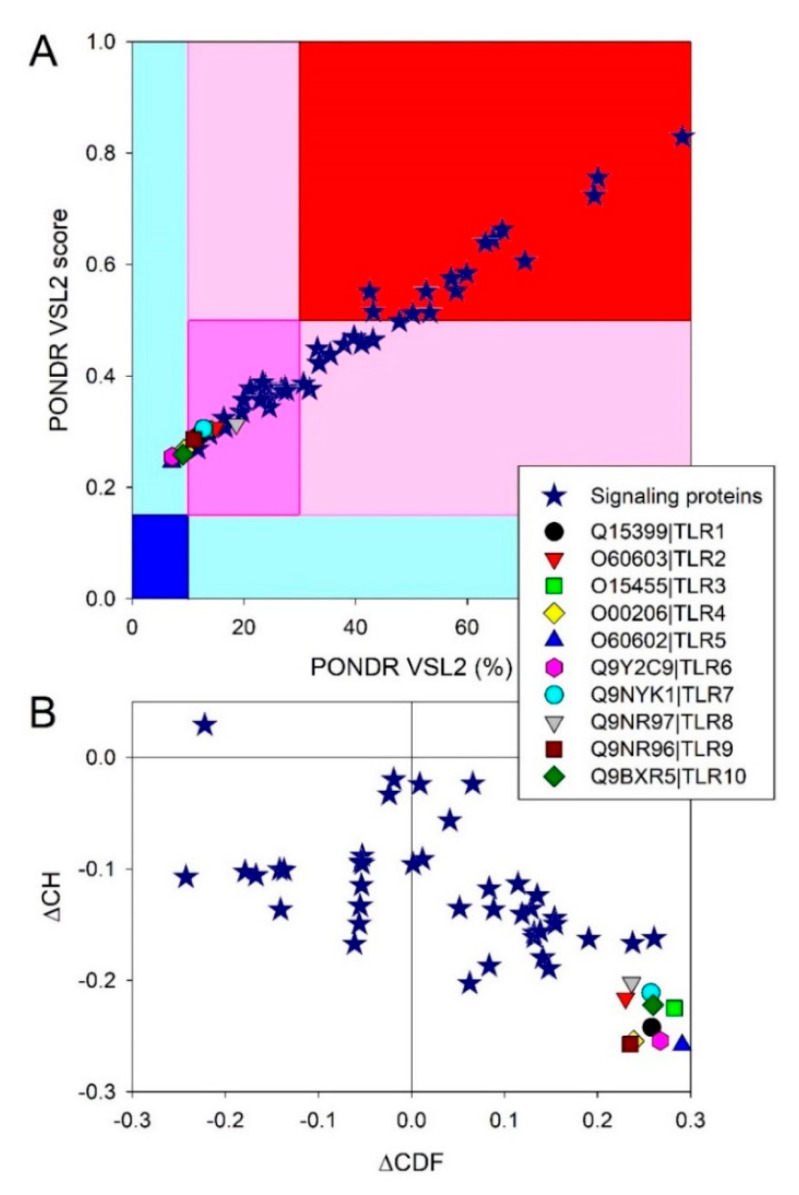
The global intrinsic disorder predisposition of human TLRs and major players in the TLR−triggered cellular pathways. (**A**) Analysis of 10 TLRs and 41 human proteins involved in TLR signaling based on the average disorder score (ADS) and percent of predicted disordered residues (PPIDR) as evaluated by PONDR^®^ VSL2 [128]. Larger values of each parameter correspond to a larger disorder propensity. Different color blocks indicate regions containing proteins with different levels of ordered, where mostly ordered, moderately disordered, and mostly disordered proteins are located within the blue, pink, and red blocks, respectively. If the two parameters (ADS and PPIDR) agree, the corresponding part of the background is shown by a dark color (blue or pink), whereas the light blue and light pink reflect areas in which only one of these criteria applies. (**B**) CH–CDF plot for 10 human TLRs and 41 major players of the TLR-based signaling. Intrinsic disorder predisposition analysis of all proteins was conducted using RIDAO web crawler, which aggregates the outputs of six well-known disorder predictors: PONDR^®^ VLXT [129], PONDR^®^ VL3 [130], PONDR^®^ VLS2 [131], PONDR^®^ FIT [132], IUPred2 (Short), and IUPred2 (Long) [133,134], and also provides the mean disorder predictions for query proteins by averaging the outputs of these six predictors. The corresponding disorder profiles of all query proteins are shown in the Appendix A. For each query protein, the predicted percentage of intrinsically disordered residues (PPIDR; i.e., percent of residues with disorder scores exceeding 0.5) was calculated based on the outputs of PONDR^®^ VLS2, which is characterized by high predictive power, as evidenced by the results of the recently conducted ‘Critical assessment of protein intrinsic disorder prediction’ (CAID) experiment, where the tool was recognized as predictor #3 of the 43 evaluated methods [135]. Global disorder status of the query proteins was checked by a CH–CDF analysis [136,137,138,139] that combines the outputs of two binary predictors, the charge–hydropathy (CH) plot [51,140] and the cumulative distribution function (CDF) plot [136,140,141], to create a CH–CDF phase space, where proteins are classified as ordered (proteins predicted to be ordered by both binary predictors), putative native “molten globules” or hybrid proteins (proteins determined to be ordered/compact by CH, but disordered by CDF), putative native coils and native pre-molten globules (proteins predicted to be disordered by both methods), and proteins predicted to be disordered by CH-plot, but ordered by CDF.

**Figure 5 biology-11-01091-f005:**
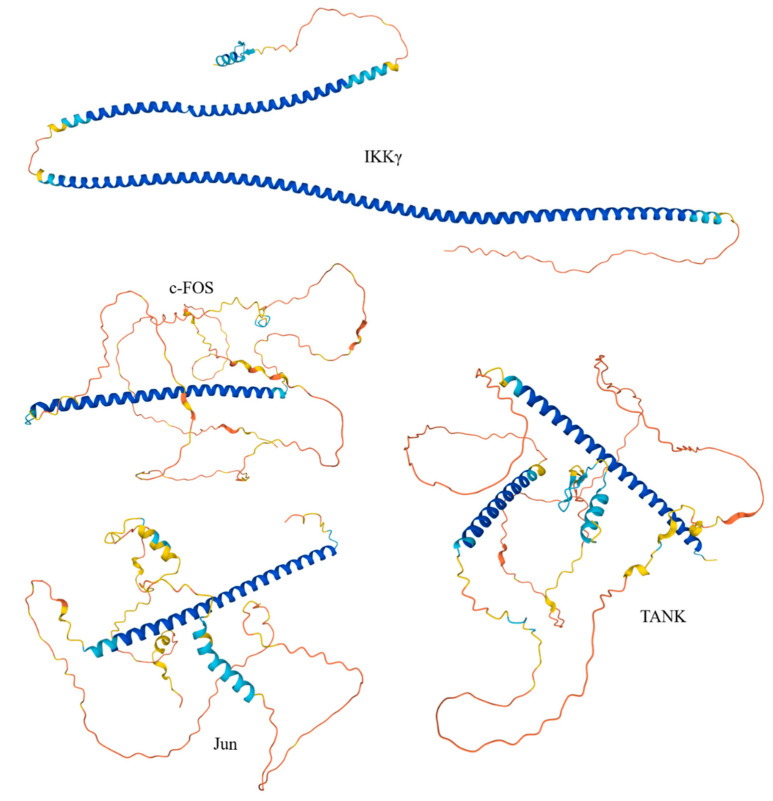
The 3D-structures modeled for human IKKγ, c-FOS, Jun, and TANK by AlphaFold [142]. Note the absence of the compact globular core in these proteins.

**Figure 6 biology-11-01091-f006:**
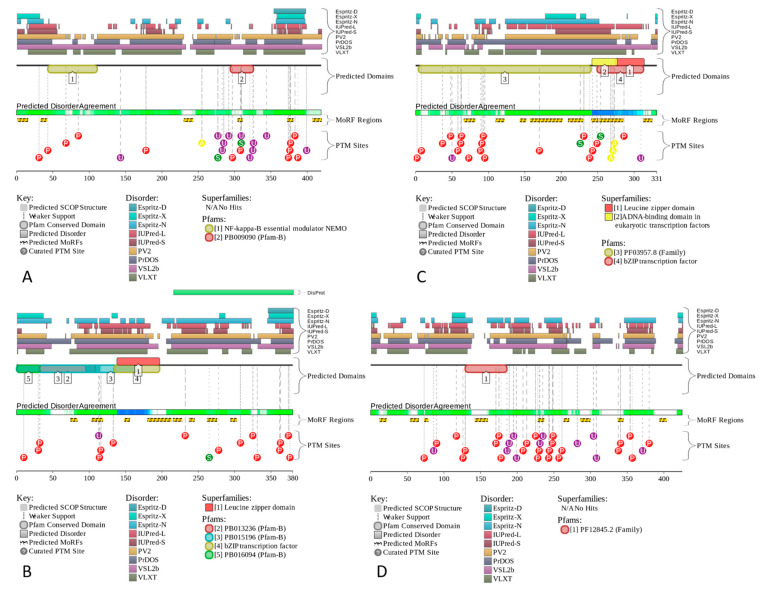
The functional disorder profiles generated for IKKγ (**A**), c-FOS (**B**), Jun (**C**), and TANK (**D**) using the D^2^P^2^ platform (http://d2p2.pro/ accessed on 7 July 2022). In addition to the outputs of nine disorder predictors, the positions of the conserved functional domains, consensus disorder score, and positions of the various PTMs, this plot also includes the locations of the predicted disorder-based binding sites (MoRF regions) identified by the ANCHOR algorithm [157] (shown by the yellow zigzagged bars).

**Figure 7 biology-11-01091-f007:**
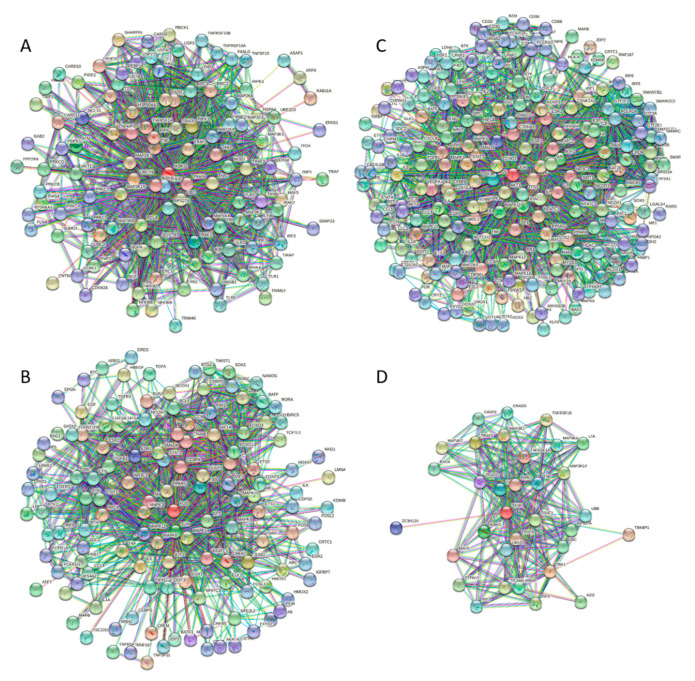
The PPI networks of human for IKKγ (**A**), c-FOS (**B**), Jun (**C**), and TANK (**D**) generated by STRING [98]. The IKKγ-centered PPI network includes 116 partners connected by 1209 interactions, indicating that each member of this network interacts with ~21 partners. c-FOS interacts with 131 proteins, each interacting with ~15 partners, thereby generating a network with 985 interactions. A total of 209 partners of Jun were linked to each other via 2076 interactions, being involved in ~20 interactions each. Finally, the TANK-centered PPI network included 52 partners connected by 320 interactions, indicating that each member of this network interacts with 12 partners.

**Figure 8 biology-11-01091-f008:**
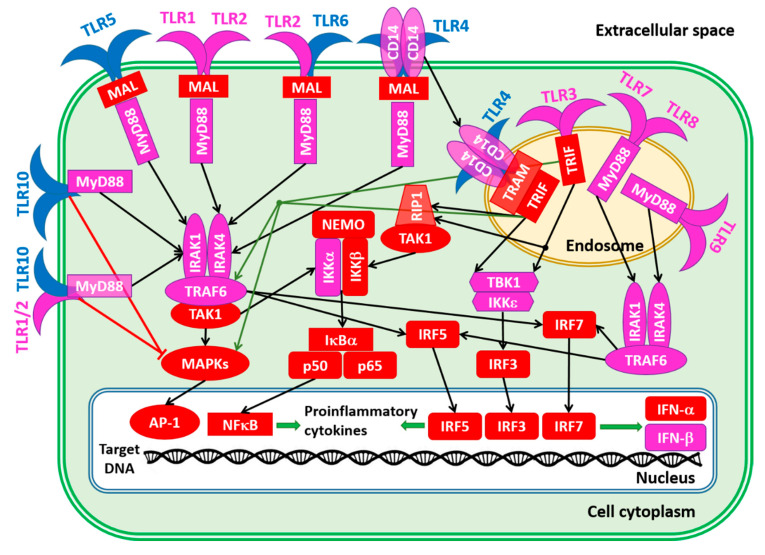
A schematic representation of the TLRs and related signaling networks.

**Figure 9 biology-11-01091-f009:**
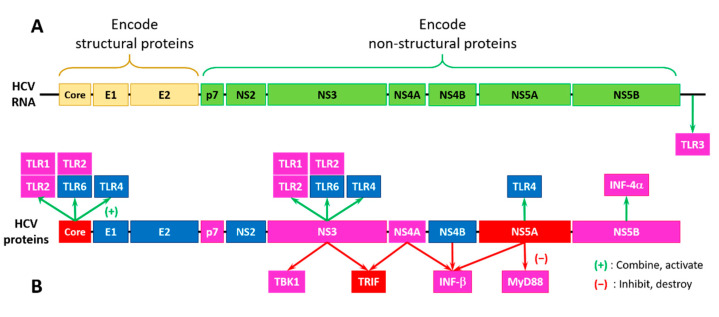
The organization of the HCV genome (**A**) and effects of the HCV-encoded proteins on Toll-like receptors (TLRs) and related downstream molecules (**B**). Proteins are colored based on their intrinsic disorder status, where highly disordered, moderately disordered, and highly ordered proteins are shown by red, pink, and blue colors, respectively. Proteins are classified based on the accepted strategy rooted in the percent of predicted intrinsic disorder (PPIDR) of query proteins, where proteins are considered as highly ordered, moderately disordered, and highly disordered, if their PPDR <10%, 10% ≤ PPDR < 30%, and PPDR ≥30%, respectively [190].

## Data Availability

The data presented in this study are available in this article and in the Appendix A.

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
