# Peer review of "Hepatitis C Virus Infection and Intrinsic Disorder in the Signaling Pathways Induced by Toll-Like Receptors"

_biology, 2022, doi:10.3390/biology11071091_

Round 1
Reviewer 1 Report
This manuscript submitted by Redwan et al uses computational analysis to define the relationship and disorderness in the proteins involved in hepatitis C virus (HCV) infection and signaling pathways. Particularly, the authors focused on the interplay between HCV proteins, toll-like receptors, and proteins act on the downstream pathways. The authors identify that all these proteins are disordered to some extent, and this could enable the protein to undergo post-translation modification (PTMs). PTMs are important regulators of protein-protein and protein-nucleic acid interactions, and thus the presence of disorder sequences could play a role in virus-host interactions.
Overall, this is well written with lots of details and proposes the role of disorderness in protein during HCV infection. The methods are well described, and figures justify the results and conclusions. Some of the minor comments are:
1. The importance of disorderness in protein has not been established well
2. The manuscript has lots of details which may need to cut, especially at the starting of “Results section”
3. The number of references are enormous in number which is hard to follow.
Author Response
Reviewer #1
This manuscript submitted by Redwan et al uses computational analysis to define the relationship and disorderness in the proteins involved in hepatitis C virus (HCV) infection and signaling pathways. Particularly, the authors focused on the interplay between HCV proteins, toll-like receptors, and proteins act on the downstream pathways. The authors identify that all these proteins are disordered to some extent, and this could enable the protein to undergo post-translation modification (PTMs). PTMs are important regulators of protein-protein and protein-nucleic acid interactions, and thus the presence of disorder sequences could play a role in virus-host interactions.
Overall, this is well written with lots of details and proposes the role of disorderness in protein during HCV infection. The methods are well described, and figures justify the results and conclusions.
RESPONSE: We are thankful to this reviewer for high valuation of our work and for constructive criticism. We tried to address all critiques and amended manuscript accordingly
Some of the minor comments are:
- The importance of disorderness in protein has not been established well
RESPONSE: The overall importance of protein intrinsic disorder is outlined in section “1.2. Introduction of the protein intrinsic disorder phenomenon” (lines 139-186). Importance of intrinsic disorder for functionality of HCV is outlined in the same section (lines 186-199). Prevalence and functionality of intrinsic disorder in TLR are described in section “3. Structure and intrinsic disorder in TLRs and major players in the TLR-triggered cellular pathways”.
- The manuscript has lots of details which may need to cut, especially at the starting of “Results section”
RESPONSE: Thank you for pointing this out. Based on this critique and as well as on the recommendations of other reviewers and Academic Editor, this manuscript was converted from Research Article to Review. We hope that this justifies the presence of various details pertaining to the TLRs and players in the TLR-triggered cellular pathways. In our view, this manuscript assembled useful information on various aspects related to TLRs, TLR-triggered cellular pathways, and roles of TLRs in HCV infection. We also hope that this collection will be of interest to the readers and scientific community.
- The number of references are enormous in number which is hard to follow.
RESPONSE: In our view, the presence of numerous references is justified by the fact that the manuscript covers multiple broad subjects and provides description of protein intrinsic disorder, HCV, TLRs, TLR-triggered cellular pathways, roles of TLRs and their pathways in HCV infection. To better reflect this, the manuscript is converted to the Review Article now.
Reviewer 2 Report
The manuscript compiled by Redwan and co-authors is indeed quite complete and interesting. However, it starts form a completely wrong assumption: this kind of manuscript is not an original research article. Any original data or novel measurements have been provided. The authors compiled a nice and complete revision of the state/of/the/art and common knowledge on HCV infection and pathophysiological conditions. Thus, we would recommend the author to edit and change the format.
At the same time, please carefully revise English language. Several statements are unclear or incorrect. I.e., line 94, “The number of genetically different subtypes depends on the genotype”. This sentence is quite redundant in terms and means almost nothing
Line 62, “Since the liver hepatocytes represent the preferential sites of the HCV replication”. There are no hepatocytes anywhere else than liver. Thus, remove liver word before hepatocytes, or opt for liver parenchymal cells as expression
Please revise and reconsider some references. Bibliography is extensive and sometimes outdated. Please cite only the relevant manuscript and, when possible, refer to previous reviews or to the most relevant original articles
Author Response
Reviewer #2
The manuscript compiled by Redwan and co-authors is indeed quite complete and interesting. However, it starts form a completely wrong assumption: this kind of manuscript is not an original research article. Any original data or novel measurements have been provided. The authors compiled a nice and complete revision of the state/of/the/art and common knowledge on HCV infection and pathophysiological conditions. Thus, we would recommend the author to edit and change the format.
RESPONSE: We are thankful to this reviewer for high valuation of our work and for constructive criticism. We tried to address all critiques and amended manuscript accordingly. Based on this critique and as well as on the recommendations of other reviewers and Academic Editor, this manuscript was converted from Research Article to Review. We hope that this justifies the presence of various details pertaining to the TLRs and players in the TLR-triggered cellular pathways.
At the same time, please carefully revise English language.
RESPONSE: Thank you for pointing this out. The manuscript was carefully proofread.
Several statements are unclear or incorrect. I.e., line 94, “The number of genetically different subtypes depends on the genotype”. This sentence is quite redundant in terms and means almost nothing
RESPONSE: Thank you for pointing this out. The corresponding part of the manuscript was changed to read: “These genotypes are further subdivided into 86 subtypes [23], which are unevenly dis-tributed between genotypes [24], and which, at the nucleotide level, show a difference ranging between 15 and 25% [22].”
Line 62, “Since the liver hepatocytes represent the preferential sites of the HCV replication”. There are no hepatocytes anywhere else than liver. Thus, remove liver word before hepatocytes, or opt for liver parenchymal cells as expression
RESPONSE: Thank you for pointing this out. This sentence was changed as recommended. It reads now: “Since the liver parenchymal cells represent the preferential sites of the HCV replication, infection with this virus causes an inflammation of the liver that leads to both acute and chronic hepatitis, which, if left untreated, often results in the development of serious lifelong illnesses such as liver fibrosis, liver cirrhosis, hepatocellular carcinoma, and end-stage liver disease in humans [3].”
Please revise and reconsider some references. Bibliography is extensive and sometimes outdated. Please cite only the relevant manuscript and, when possible, refer to previous reviews or to the most relevant original articles
RESPONSE: In our view, the presence of numerous references is justified by the fact that the manuscript covers multiple broad subjects and provides description of protein intrinsic disorder, HCV, TLRs, TLR-triggered cellular pathways, roles of TLRs and their pathways in HCV infection. To better reflect this, the manuscript is converted to the Review Article now. Please note that of 231 references in this review, only 17 were published before 2000, and 60% (almost 140) of all cited papers were published between 2010 and 2022.
Reviewer 3 Report
Observations of authors are in line with findings in other virus-associated human diseases such as human papillomaviruses, influenza viruses, HIV-1, Dengue virus, respiratory syncytial virus, Zika virus, Chikungunya virus, rotavirus, Japanese encephalitis virus, SARS-CoV-2 etc. REsearch should be focused on further investigations leading to successful broad-spectrum anti-viral therapy.
Author Response
Reviewer #3
Observations of authors are in line with findings in other virus-associated human diseases such as human papillomaviruses, influenza viruses, HIV-1, Dengue virus, respiratory syncytial virus, Zika virus, Chikungunya virus, rotavirus, Japanese encephalitis virus, SARS-CoV-2 etc. REsearch should be focused on further investigations leading to successful broad-spectrum anti-viral therapy.
RESPONSE: We are thankful to this reviewer for high valuation of our work.
Reviewer 4 Report
In this manuscript, the authors examined the interplay between HCV and TLR signaling pathways by in silico analysis.
1. The manuscript is very descriptive, and the biological significance is unclear.
2. It's unclear how to apply the results to medicine. Are they related to the development of novel therapy against HCV hepatitis?
3. The novelty of this manuscript is unclear.
4. No biological experiments were performed.
Author Response
Reviewer #4
In this manuscript, the authors examined the interplay between HCV and TLR signaling pathways by in silico analysis.
- The manuscript is very descriptive, and the biological significance is unclear.
RESPONSE: We are thankful to this reviewer for high valuation of our work and for constructive criticism. We tried to address all critiques and amended manuscript accordingly. Based on this critique and as well as on the recommendations of other reviewers and Academic Editor, this manuscript was converted from Research Article to Review. We hope that this justifies the presence of various details pertaining to the NCV, TLRs, and players in the TLR-triggered cellular pathways.
- It's unclear how to apply the results to medicine. Are they related to the development of novel therapy against HCV hepatitis?
RESPONSE: Thank you for pointing this out. The corresponding clarification is added to the conclusion section.
- The novelty of this manuscript is unclear.
RESPONSE: Thank you for pointing this out. The corresponding clarification is added to the conclusion section.
- No biological experiments were performed.
RESPONSE: Thank you for pointing this out. As we already indicated, based on the recommendations of other reviewers and Academic Editor, this manuscript was converted from Research Article to Review. Therefore, no biological experiments are required.
Round 2
Reviewer 2 Report
The manuscript submitted by Redwan and co-authors offers a nice overview of hepatitis C infection in relation to toll-like receptors (and downstream proteins). The R2 manuscript has been properly revised and edited according to revisors’ requests. The title is still quite misleading: the manuscript describes the interaction between the HCV virus and TLRs. In particular, the authors highlight TLR induction in response to the virus. Thus the title may better sound as “Hepatitis C infection and pathogenesis in relation to toll-like receptors” or even better “Toll-Like Receptors and related proteins in Hepatitis C infection and pathogenesis”
Overall, the manuscript is ready for publication as a review manuscript in Biology.
Reviewer 4 Report
The authors changed the manuscript from Research Article to Review. As a review article, the authors summarize TLRs, HCV and their association very well.